

**The response of precipitation characteristics to global warming from global**
**and regional climate projections**
*Filippo Giorgi, Francesca Raffaele, Erika Coppola*
*1 Earth System Physics Section, The Abdus Salam International Centre for Theoretical*
*Physics, I-34151 Trieste, Italy*
Corresponding author: Filippo Giorgi, Earth System Physics Section, The Abdus Salam
International Centre for Theoretical Physics, Trieste, I-34151, Italy. Email: giorgi@ictp.it;
phone: +39 0402240425.
Submitted to: Earth System Dynamics



**Abstract**

We revisit the issue of the response of the precipitation characteristics to global warming
based on analyses of global and regional climate model projections for the 21st century. The
prevailing response we identify can be summarized as follows: increase in the intensity of
precipitation events and extremes, with the occurrence of events of "unprecedented"
magnitude, i.e. magnitude not found in present day climate; decrease in the number of light
precipitation events and in wet spell lengths; increase in the number of dry days and dry spell
lengths. This response, which is mostly consistent across the models we analized, is tied to the
difference between precipitation intensity responding to increases in local humidity
conditions, especially for heavy and extreme events, and mean precipitation responding to
slower increases in global evaporation. These changes in hydroclimatic characteristics have
multiple and important impacts on the Earth's hydrologic cycle and on a variety of sectors,
and as examples we investigate effects on the potential stress due to increases in dry and wet
extremes, changes in precipitation interannual variability and changes in potential
predictability of precipitation events. We also stress how the understanding of the
hydroclimatic response to global warming can shed important insights into the fundamental
behavior of precipitation processes, most noticeably tropical convection.
**Keywords**: Precipitation, climate change, hydrologic cycle, extremes

## 1. Introduction

One of the greatest concerns regarding the effects of climate change on human
societies and natural ecosystems is the response of the Earth's hydrologic cycle to global
warming. In fact, by affecting the surface energy budget, greenhouse gas (GHG) induced
warming, along with related feedback processes (e.g. the water vapor, ice albedo and cloud



feedbacks), can profoundly affect the Earth's water cycle (e.g. Trenberth et al. 2003; Held and
Soden 2006; Trenberth 2011; IPCC 2012).

The main engine for the Earth's hydrologic cycle is the radiation from the Sun, which

heats the surface and causes evaporation from the oceans and land. Total surface evaporation
has been estimated at 486 $10^3$ km$^3$/year of water, of which 413 $10^3$ km$^3$/year, or ~85%, is
from the oceans and the rest from land areas (Trenberth et al. 2007). Once in the atmosphere,
water vapor is transported by the winds until it eventually condenses and forms clouds and
precipitation. The typical atmospheric lifetime of water vapor is of several days, and therefore
at climate time scales there is essentially an equilibrium between global surface evaporation
and precipitation. Total mean precipitation as been estimated at 373 $10^3$ km$^3$/year of water
over oceans and 113 $10^3$ km$^3$/year over land (adding up to the same global value as
evaporation, Trenberth et al. 2007). Water precipitating over land can then either re-evaporate
or flow into the oceans through surface runoff or sub-surface flow.

Given this picture of the hydrologic cycle, however, it is important to stress that,

although evaporation and precipitation globally balance out, their underlying processes are
very different. Evaporation is a continuous and slow process (globally about ~2.8 mm/day,
Trenberth et al. 2007), while precipitation is a highly intermittent, fast and localized
phenomenon, with precipitation events drawing moisture only from an area of about 3-5 times
the size of the event itself (Trenberth et al. 2003). In addition, on average, only about 25% of
days are rainy days (where throughout this paper a rainy day is considered has having a
precipitation amount of at least 1 mm/day, so that drizzle days are removed), but since it does
not rain throughout the entire day, the actual fraction of time it rains has been estimated at 5-
10% (Trenberth et al. 2003). In other words, most of the time it does not actually rain.



This has important implications for the assessment of hydroclimatic responses to
global warming, because it may not be very meanigful, and certainly not sufficient, to analyze
mean precipitation fields, but it is necessary to also investigate higher order statistics. For
example, the same mean of, say, 1 mm/day could derive from 10 consecutive 1 mm/day
events, a single 10 mm/day event with 9 dry days, or two 5 mm/day events separated by a dry
period. Each of these cases would have a very different impact on societal sectors or
ecosystem dynamics.
This consideration also implies that the impact of global warming on the Earth's
hydroclimate might actually manifest itself not only as a change in mean precipitation but,
perhaps more markedly, as variations in the characteristics and regimes of precipitation
events. This notion has been increasingly recognized since the pioneering works of Trenberth
(1999) and Trenberth et al. (2003), with many studies looking in particular at changes in the
frequency and intensity of extreme precipitation events (e.g. Easterling et al. 2000;
Christensen and Christensen 2003; Tebaldi et al. 2006; Allan and Soden 2008; Giorgi et al.
2011; IPCC 2012; Sillmann et al. 2013; Giorgi et al. 2014a,b).
In this paper we revisit some of the concepts related to the issue of the impacts of
global warming on the characteristics of the Earth's hydroclimate, stressing however that it is
not our purpose to provide a review of the extensive literature on this topic. Rather, we want
to illustrate some of the points made above through relevant examples obtained from new and
past analyses of global and regional climate model projections.
More specifically, we will draw from global climate model (GCM) projections carried
out as part of the CMIP5 program (Taylor et al. 2012) and regional climate model (RCM)
projections from the COordinated Regional climate Downscaling EXperiment (CORDEX,
Giorgi et al. 2009; Jones et al. 2011, Gutowski et al. 2016), which downscale CMIP5 GCM



data. In this regard, we focus on the high end RCP8.5 scenario, in which the ensemble mean
global temperature increase by 2100 is about 4°C (+/- 1°C) compared to late 20th century
temperatures (IPCC 2013), stressing that results for lower GHG scenarios are qualitatively
similar to those found here but of smaller magnitude (not shown for brevity).

In the next sections we first summarize the changes in mean precipitation fields in our

ensemble of model projections, and then explore the response of different precipitation
characteristics, trying specifically to identify robust responses. After having identified the
dominant hydroclimatic responses, we discuss examples of their impact on different quantities
of relevance for socio-economic impacts, and specifically the potential stress associated with
changes in dry and wet extreme events, precipitation interannual variability and predictability
of precipitation events.
**2. The hydroclimatic response to global warming**
***2.1 Mean precipitation changes***

In general, as a result of the warming of the oceans and land, global surface

evaporation increases with increasing GHG forcing. This increase mostly lies in the range of
1-2 % per degree of surface global warming (%/DGW; Trenberth et al. 2007). As a
consequence, global mean precipitation also tends to increase roughly by the same amount.
This has been found in most GCM projections, as illustrated in the examples of Figure 1.



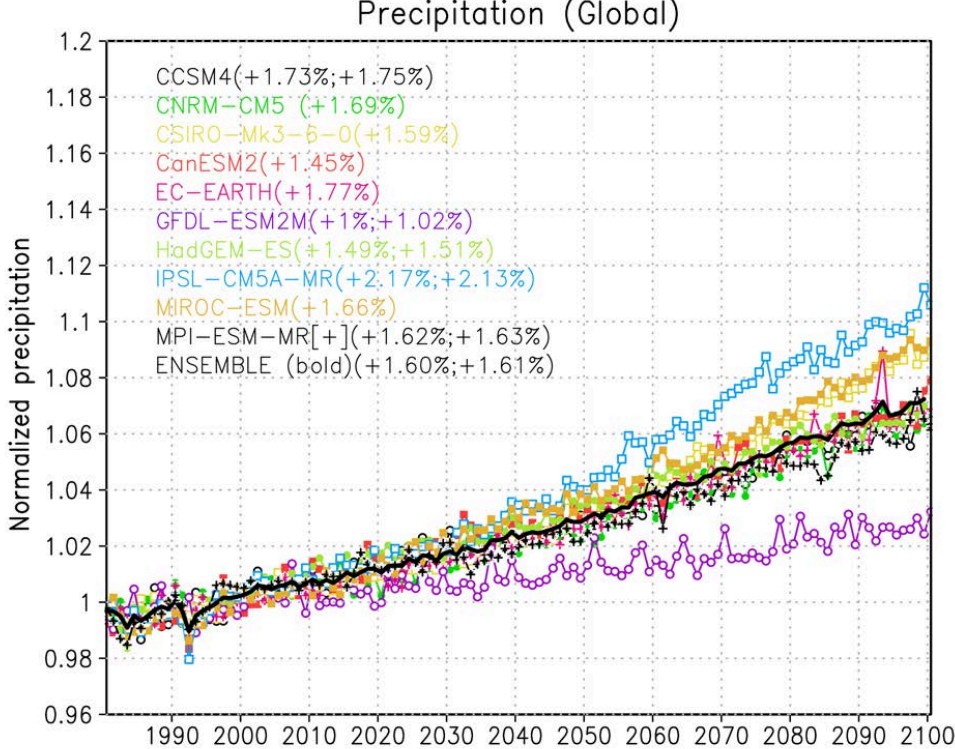

**Figure 1.** Normalized mean global precipitation from 1981 to 2100 in the 10 CMIP5 GCMs simulation for the RCP8.5 scenario used by Giorgi et al. (2014b), along with their ensemble average. The first number in parentheses shows the corresponding mean global precipitation change per degree of global warming, while the second shows (for a subset of models with available data) the same quantity for global surface evaporation. The annual precipitation is normalized by the mean precipitation during the reference period 1981-2010, therefore a value of, e.g., 1.1 indicates an increase of 10%.

Although precipitation increases globally, at the regional level we can find relatively complex patterns of change, with areas of increased and areas of decreased precipitation. These patterns are closely related to changes in global circulation features, local forcings (e.g. topography, land use) and energy and water fluxes affecting convective activity. The basic geographical structure of precipitation change patterns has been quite resilient throughout different generations of GCM projections, at least in an ensemble averaged sense. These



precipitation change patterns are shown in Figure 2 as obtained from the CMIP5 ensemble,
but they are similar in the CMIP3 and earlier GCM ensembles.

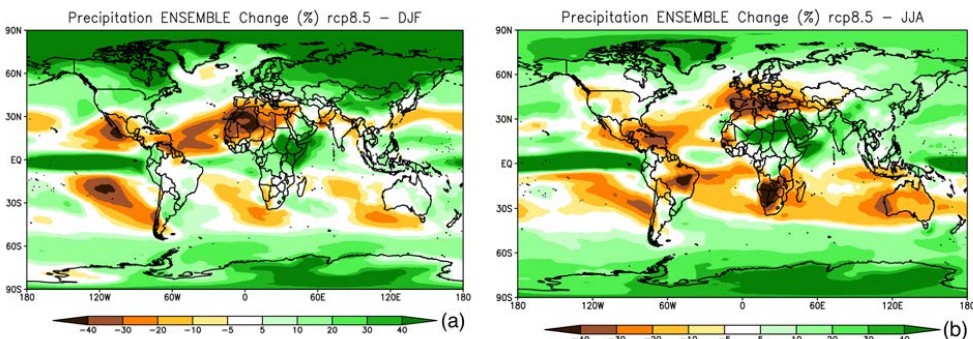

**Figure 2.** Ensemble mean change in precipitation (RCP8.5, 2071-2100 minus 1981-2010) for
December-January-February (panel a) and June-July-August (panel b) in the CMIP5 ensemble of models.
The patterns of Figure 2 have been often referred to as "the rich get richer and the poor
get poorer" in the sense that mid and high latitude regions, the Intertropical Convergence
Zone (ITCZ) and some tropical monsoon regions, which are already wet in present climate
conditions, are projected to become wetter with global warming, while dry sub-tropical
regions are projected to become drier.
The increase in precipitation at mid to high latitudes has been attributed to a poleward
shift of the storm tracks associated with maximum warming in the tropical troposphere (due
to enhanced convection), which in turn produces a poleward shift of the maximum horizontal
temperature gradient and jet stream location (e.g. IPCC 2013). This process is essentially
equivalent to a poleward expansion of the Hadley Cell, which also causes drier conditions in
sub-tropical areas, including the Mediterranean and Central America/Southwestern U.S.
regions. Conversely, the increase in precipitation over the ITCZ is due to increased
evaporation over the equatorial oceans, which feeds and intensifies local convective systems.
Finally, over monsoon regions, a general increase of precipitation has been attributed to the



greater water-holding capacity of the atmosphere that counterbalances a decrease in monsoon
circulation strength (IPCC 2013).

As already mentioned, these broad scale change patterns have been confirmed by

different generations of GCM projections, and thus appear to be robust model-derived signals.
On the other hand, high resolution RCM experiments have shown that local forcings
associated with complex topography and coastlines can substantially modulate these large
scale signals, often to the point of being of opposite sign. For example, the precipitation
shadowing effect of major mountain systems tends to concentrate precipitation increases
towards the upwind side of the mountains, and to reduce the increases or even generate
decreases of precipitation in the lee side (e.g. Giorgi et al. 1994; Gao et al. 2006). Similarly, in
the summer, the precipitation change signal can be strongly affected by high elevation
warming and wetting which enhance local convective activity. For example, Giorgi et al.
(2016) found enhanced precipitation over the Alpine high peaks in high resolution EURO-
CORDEX (Jacob et al. 2014) and MED-CORDEX (Ruti et al. 2016) projections, whereas the
driving coarse resolution global models produced a decrease in precipitation. In addition to
these local effects, it has been found that the simulation of some modes of variability, such as
blocking events, is also sensitive to model resolution (e.g. Anstey et al. 2013, Schiemann et al.
2017). As a result of all these processes it is thus possible that the "rich get richer, poor get
poorer" patterns might be significantly modified as we move to substantially higher resolution
models.

On the other hand, the question could be posed: "How richer will the rich get and how

poorer will the poor get?". This question depends more on the modifications of the
characteristics of precipitation than the mean precipitation itself. For example, changes in
precipitation interannual variability may have strong impacts on crop planning. As another
example, if an increase in precipitation is due to an increase of extreme damaging events, this



will have negative rather than positive impacts. Alternatively, if the increase is due to very
light events that do not replenish the soil of moisture, this will not constitute an added water
resource. Conversely, if a reduction of precipitation is mostly associated with a reduction of
extremes, then this will result in positive impacts. It is thus critical to assess how the
characteristics of precipitation will respond to global warming, which is the focus of the next
sections.
*2.2 Daily precipitation intensity Probability Density Functions (PDFs)*
Daily precipitation is one of the variables most often used in impact assessment
studies, therefore an effective way to investigate the response of precipitation characterstics to
global warming is to assess changes in daily precipitation intensity PDFs. As an illustrative
example of PDF changes, Figures 3 and 4 show normalized precipitation intensity PDFs for 4
time slices, 1981-2010 (reference period representative of present day conditions), 2011-2040,
2041-2070 and 2071-2100 in the MPI-ESM-MR RCP8.5 projection of the CMIP5 ensemble.
The farther the time slice is in the future, the greater the warming (up to a maximum of about
4 °C in 2071-2100). The variable shown, which we refer to as PDF, is the frequency of
occurrence of precipitation events within a certain interval (bin) of intensity normalized by the
total number of days, including non-precipitating days, where a day is considered to be rainy
if the daily precipitation intensity is above 1 mm/day (as in Giorgi et al. 2014b). Given the
logarithmic scale of the frequency of occurrence, in order to better illustrate changes in
frequencies, the figure reports the ratio of the frequency of occurrence for a given bin in a
future time slice divided by the same quantity in the reference period. Finally, averaged data
are shown for land areas in the tropics (30°S-30°N, Figure 3), and extra-tropical midlatitudes
(30-60° N and S, Figure 4), noting that qualitatively similar results were found for ocean
areas.



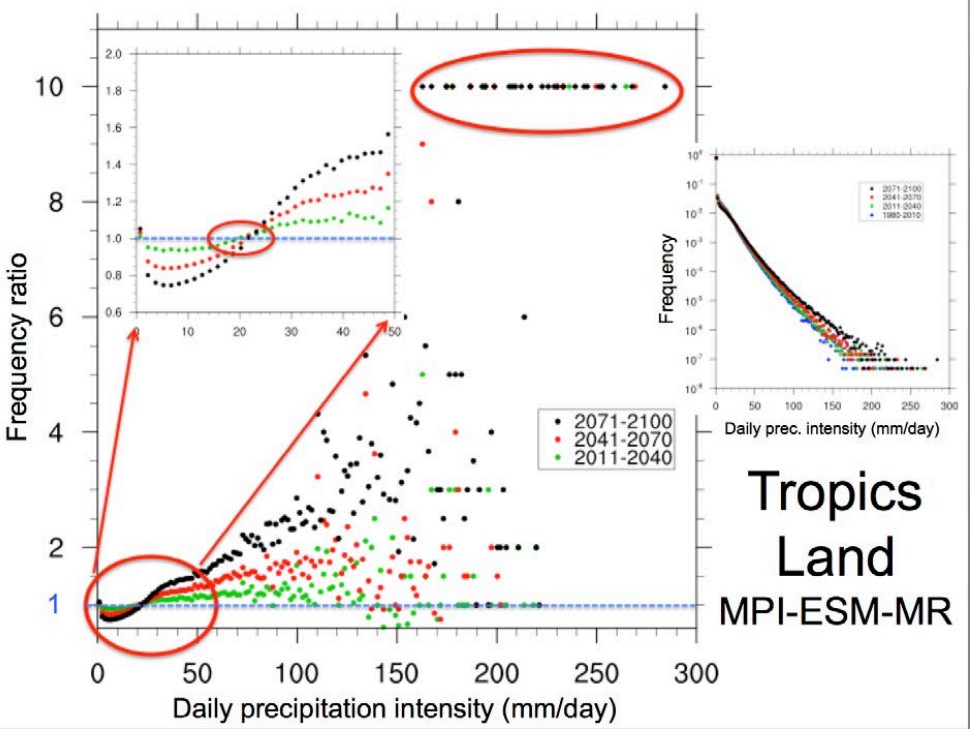

**Figure 3.** Small right panel: Probability density function (PDF) defined as the normalized frequency of occurrence of daily precipitation events of intensity within a certain bin interval over land regions in the tropics (30°S - 30°N) for the reference period 1981-2010 and three future time slices (2011-2040, 2041-2070, 2071-2100) in the MPI-ESM-MR model. The frequency is normalized by the total number of days (including dry days, i.e. days with precipitation lower than 1 mm/day). Large central panel: Ratio of future to reference normalized frequency of daily precipitation intensity for the three future time slices. The small inset panel shows a zoom on the part of the curves highlighted by the corresponding red oval. Ratio values of 10 (hilighted in a red oval) are used when events occur in the future time slice which are not present in the reference period for a given intensity bin.

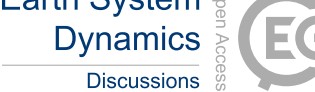



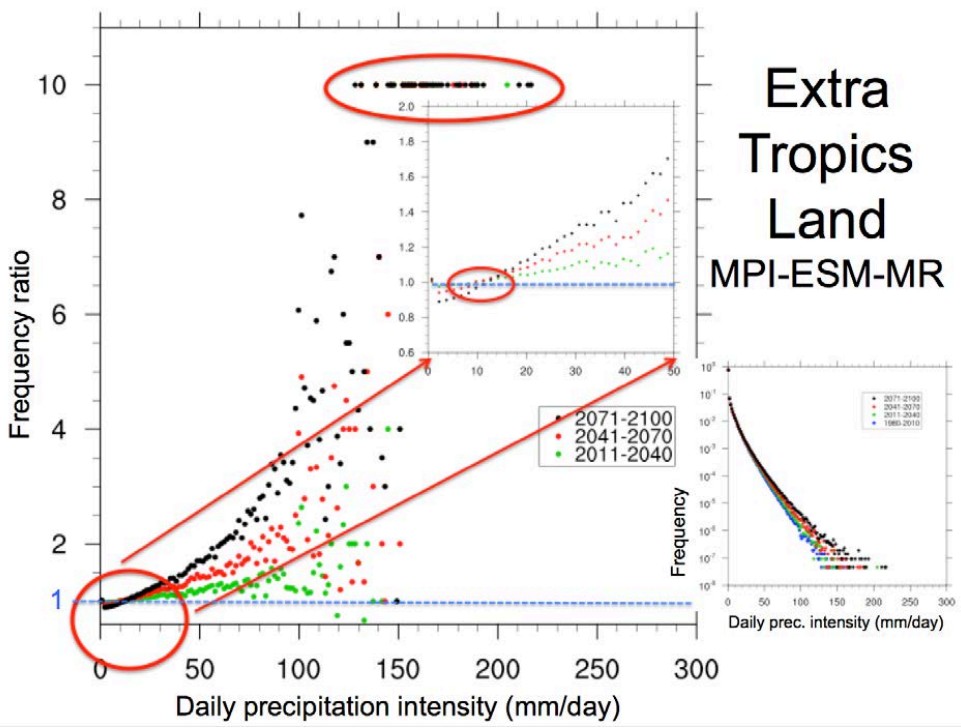

**Figure 4.** Same as Figure 3 but for extra-tropical land areas.

The PDFs exhibit a log-linear relationship between intensities and frequencies, with a sharp drop in frequency as the intensity increases. The ratios of future vs. present day frequencies consistently show the following features:

i) An increase in the number of dry days, as seen from the ratios > 1 in the first bin (precipitation less than 1 mm/day), i.e. a decrease in the frequency of wet events. Note that, even if these ratios are only slightly greater than 1, because the frequencies of dry days are much higher than those of wet days, the actual absolute increase in the number of dry days is relatively high.

ii) A decrease (ratio < 1) in the frequency of light to medium precipitation events up to a certain intensity threshold. In the models we analyzed, when taken over large areas, this



threshold lies around the 95th percentile of the full distribution, and is higher for tropical than
extratropical land regions because of the higher amounts of precipitation in tropical
convection systems. Interestingly, while the threshold depends on latitude, it is approximately
invariant for all future time slices, i.e. it appears to be relatively independent of the level of
warming. The decrease in light precipitation events has been at least partially attributed to an
increase in thermal stability induced by the GHG forcing (Chou et al. 2012).

iii) An increase (ratio > 1) in the frequency of events for intensities higher than the

threshold mentioned above. The relative increase in frequency grows with the intensity of the
events, and it is thus maximum for the highest intensity events, an indication of a non linear
response of the precipitation intensity to warmer conditions. Note that, because of the
logarithmic frequency scale, the absolute increase in the number of high intensity events is
relatively low.

iv) The occurrence in the future time slices of events with intensity well beyond the

maximum found in the reference period. These are illustrated by the prescribed value of 10
when events occurred for a given bin in the future time slice, but not in the reference one. One
could thus interpret these as occurrences of "unprecedented" events.

v) All the features i)-iv) tend to amplify as the time slice is further into the future, i.e.

as the level of warming increases, and are generally more pronounced over tropical than
extratropical areas (and over land than ocean regions, which we did not show for brevity).

Although the results in Figures 3 and 4 are obtained from one model, they are

qualitatively consistent with those we found for other CMIP5 GCMs (not shown for brevity).
We also carried out the same type of analysis for a high resolution RCM projection (12 km
grid spacing, RCP8.5 scenario) conducted with the RegCM4 model (Giorgi et al. 2012) over
the Mediterranean domain defined for the MED-CORDEX program (Ruti et al. 2016).
Figures 5 and 6 show PDFs and PDF ratios for three 30-year future time slices calculated over



land areas throughout the Mediterranean domain and over a sub-area covering the Alpine
region. They show features similar to those found for the GCMs, with the signal over the
Alpine region being more pronounced than for the entire Mediterranean area. In addition, our
results are qualitatively in line with previous analyses of RCM projections (e.g. Gutowski et
al. 2007; Boberg et al. 2009; Jacob et al 2014; Giorgi et al. 2014a), suggesting that the
projected changes in precipitation intensity PDFs summarized in the points i)-iv) above are
generally robust across a wide range of models and model resolutions.

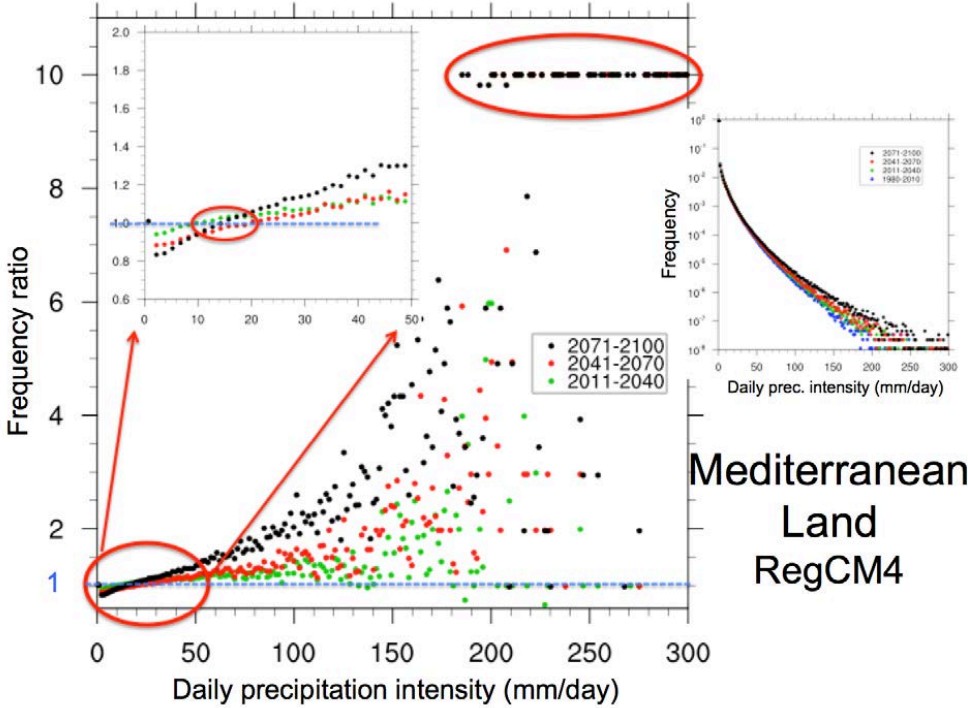


**Figure 5.** Same as Figure 3 but for Mediterranean land areas in a MED-CORDEX experiment with the

RegCM4 RCM driven by global fields from the HadGEM GCM.





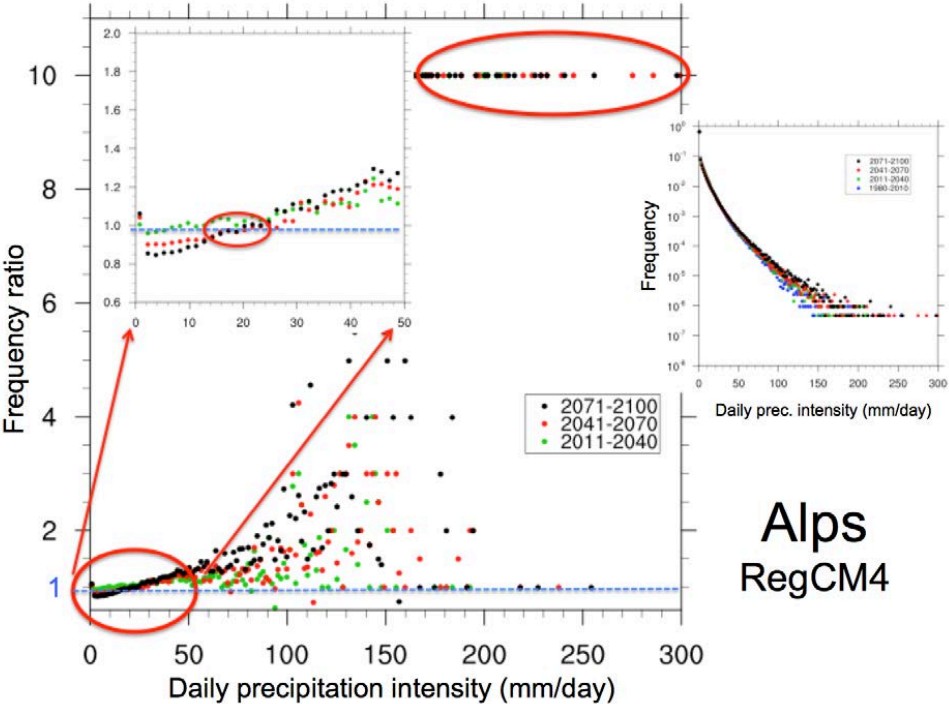

**Figure 6.** Same as Figure 5 but for the Alpine region.

### 2.3 Hydroclimatic indices

The changes in precipitation intensity PDFs found in the previous section should be reflected in, and measured by, changes of hydroclimatic indices representative of given precipitation regimes. In two previous studies (Giorgi et al. 2011, 2014b), we assessed the changes of a series of interconnected hydroclimatic indices in an ensemble of 10 CMIP5 projections. The indices analyzed include:

SDII: Mean precipitation intensity (including only wet events)

DSL: Mean dry spell length, i.e. mean length of consecutive dry days

WSL: Mean wet spell length, i.e. mean length of consecutive wet days



R95: Fraction of total precipitation above the 95th percentile of the daily precipitation
intensity distribution during the reference period 1981-2010.
PA: Precipitation area, i.e. the total area covered by wet events at any given day
HY-INT, i.e. the hydroclimatic intensity index introduced by Giorgi et al. (2011)
consisting of the product of normalized SDII and MDSL.
Note that the PA and HY-INT indices were specifically introduced by Giorgi et al.
(2011, 2014b). The PA is the spatial counterpart of the mean frequency of precipitation days,
while the HY-INT was introduced under the assumption that the changes in SDII and MDSL
are interconnected responses to global warming (Giorgi et al. 2011).
Giorgi et al. (2011, 2014b) examined changes in these indices for ensembles of
CMIP3 and CMIP5 GCM projections, as well as a number of RCM projections, in future time
slices with respect to the 1976-2005 reference period. Their results, which were consistently
found for most models analyzed, are schematically depicted in Figure 7, which shows that
under warming conditions the models indicate a prevalent increase in SDII, R95, HY-INT and
DSL and a decrease in PA and WSL. Similar results where then found by Giorgi et al. (2014a)
in an analysis of multiple RegCM4-based projections over 5 CORDEX domains. In other
words, under warmer climate conditions, precipitation events are expected to be more intense
and extreme and temporally more concentrated and less frequent, which implies a reduction
of the areas occupied by rain at any given time (although not necessarily a reduction of the
size of the events). This conclusion is consistent with the change in PDFs illustrated in
Figures 3-6.





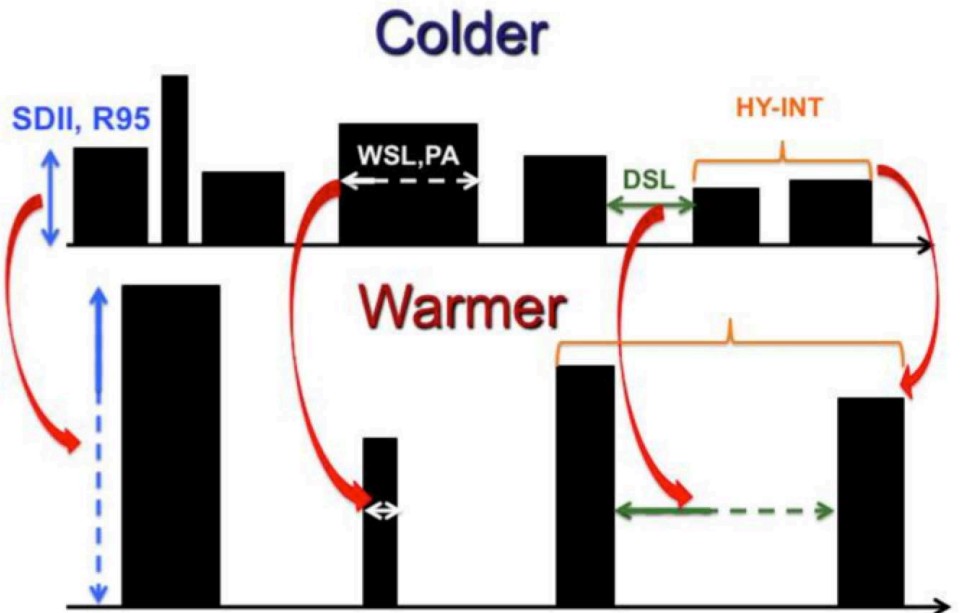


**Figure 7.** Schematic depiction of the hydroclimatic response to climate warming emerging from the
analysis of multiple indices by Giorgi et al. (2014b), which showed an increase in SDII, R95, HY-INT and DSL
and a decrease in PA and WSL. Each box represents a precipitation event whose area is the total amount of
precipitaiton, the height is the intensity and the horizontal length the duration. The interval between two events
represents a dry period.
In addition, Giorgi et al. (2011 and 2014b) analyzed a global and several regional daily
precipitation gridded observation datasets, and found that trends for the period 1976-2005
were predominantly in line with the changes illustrated in Figure 7 over most continental
areas. Further evidence of increases in heavy precipitation events in observational records is
for example reported by Fischer and Knutti (2016) and references therein, however this
conclusion cannot be considered entirely robust, and needs to be verified with further
analysis, due to the high uncertainty in precipitation observations (e.g. Herold et al. 2017).
An explanation for the hydroclimatic response to global warming illustrated in Figure
7 is related to the fact that, on the one hand, the mean global precipitation change roughly



follows the mean global evaporation increase, i.e. 1.5-2.0 %/DGW (Trenberth et al. 2007,
Figure 1), while, on the other hand the intensity of precipitation, in particular for high and
extreme precipitation events, is more tied to the increase in the water holding capacity of the
atmosphere, which is in turn regulated by the Clausius-Clapeyron (Cl-Cl) response of about
7%/DGW (e.g. Trenberth et al. 2003; Pall et al. 2007; Lenderink and van Meijgaard 2008;
Chou et al. 2012; Singleton and Toumi 2013; Ivancic and Shaw 2016; Fischer and Knutti
2016). Therefore the increase in precipitation intensity can be expected to be larger than the
increase in mean precipitation, which implies a decrease in precipitation frequency.
To illustrate this point, Table 1 reports the globally averaged changes (2071-2100
minus the reference period 1976-2005, as in Giorgi et al. 2014b; RCP8.5 scenario) in mean
precipitation, precipitation intensity and frequency, and 95th, 99th and 99.9th percentile of
daily precipitation for the 10 GCMs of Giorgi et al. (2014b), along with their ensemble
average. The values of Table 1 were calculated as follows: we first computed the change in
%/DGW at each model grid point and then averaged these values over global land+ocean as
well as global land-only areas. This was done in order to avoid the possiblity that areas with
large precipitation amounts may dominate the average. On the other hand, grid-point
normalization artificially amplifies the contribution of regions with small precipitation
amounts, such as polar and desert areas. For this reason, as in Giorgi et al. (2014), we did not
include in the averaging areas north of 60$^o$N and south of 60 $^o$S (polar regions) along with
areas with mean annual precipitation lower than 0.5 mm/day (which effectively identifies
desert regions). In addition, we did not consider precipitation associated with days with
amounts of less than 1 mm/day in order to be consistent with our definition of rainy day
(which disregards drizzle events).



| Global Box | | | | | |
|---|---|---|---|---|---|
| Models | N. Wet Days %/ DGW | Precipitation change (due to wet days)%/ DGW | SDII change(%)/ DGW | 95p change(%)/ DGW | 99p change(%)/ DGW | 99.9p change(%)/ DGW |
| HadGEM-ES | -0.7 | 1.3 | 1.8 | 1.7 | 2.9 | 3.9 |
| MPI-ESM-MR | -2.4 | 1.0 | 3.5 | 1.9 | 3.7 | 5.3 |
| GFDL-ESM2M | -1.4 | 0.05 | 1.2 | 0.3 | 2.1 | 10.4 |
| IPSL-CM5A-MR | -1.0 | 1.6 | 2.6 | 2.0 | 4.5 | 7.9 |
| CCSM4 | -1.1 | 0.7 | 1.8 | 1.1 | 2.8 | 5.5 |
| CanESM2 | -0.4 | 1.6 | 1.7 | 1.5 | 2.5 | 4.4 |
| EC-EARTH | -0.9 | 1.3 | 2.1 | 1.9 | 3.7 | 5.9 |
| MIROC-ESM | 0.2 | 1.4 | 0.9 | 1.1 | 1.2 | 1.6 |
| CSIRO-Mk3-6-0 | -0.6 | 0.8 | 1.9 | 2.3 | 2.4 | 3.4 |
| CNRM-CM5 | -0.1 | 1.4 | 1.5 | 1.5 | 2.9 | 5.8 |
| **ENSEMBLE** | **-0.8** | **1.1** | **1.9** | **1.5** | **2.9** | **5.4** |

| Global LAND Box | | | | | |
|---|---|---|---|---|---|
| Models | N. Wet Days %/ DGW | Precipitation change (due to wet days)%/ DGW | SDII change(%)/ DGW | 95p change(%)/ DGW | 99p change(%)/ DGW | 99.9p change(%)/ DGW |
| HadGEM-ES | -1.4 | 0.7 | 2.1 | 1.2 | 2.8 | 4.5 |
| MPI-ESM-MR | -3.3 | 0.1 | 4.0 | 0.8 | 3.7 | 5.4 |
| GFDL-ESM2M | -1.8 | 1.1 | 3.1 | 1.2 | 4.5 | 12.4 |
| IPSL-CM5A-MR | -1.8 | 0.7 | 2.5 | 1.2 | 3.8 | 7.2 |
| CCSM4 | -0.6 | 1.3 | 1.9 | 1.3 | 2.8 | 5.4 |
| CanESM2 | -0.6 | 1.2 | 1.7 | 1.3 | 3.4 | 5.0 |
| EC-EARTH | -0.8 | 1.4 | 2.3 | 2.0 | 3.8 | 6.0 |
| MIROC-ESM | 0.2 | 1.8 | 1.4 | 1.1 | 1.7 | 2.1 |
| CSIRO-Mk3-6-0 | -1.8 | -0.2 | 1.5 | 0.2 | 1.1 | 2.4 |
| CNRM-CM5 | 0.4 | 2.5 | 2.0 | 2.0 | 3.2 | 6.0 |
| **ENSEMBLE** | **-1.2** | **1.1** | **2.3** | **1.2** | **3.1** | **5.6** |

**Table 1.** Change in different daily precipitation indicators between 2071-2100 and 1976-2005 for the 10
CMIP5 GCMs of Giorgi et al. (2014b) expressed in % per degree of surface global warming over global (upper
box) and global-land (lower box) areas, where global means the area between 60°S and 60°N. SDII is the
precipitaiton intensity, 95p, 99p and 99.9p are the 95th, 99th and 99.9th percentiles, respectively,  and the
precipitation change only include wet days, i.e. days with precipitation greater than 1 mm/day.

Also in these calculations, the increase in global mean precipitation is in the range of

1-2 %/DGW except for the GFDL experiment, which shows a very small increase (indicating
that in this model most of the precipitation increase occurs in the polar regions). In all cases
except for MIROC the increase in global SDII is greater than the increase in mean



precipitation, resulting in a decrease of the number of rainy days. The changes in the 95th,
99th and 99.9th percentile are maximum for the most extreme percentiles, showing that the
main contribution to the response of Figure 7 is due to the highest intensity events, i.e. above
the 99th and 99.9th percentiles, whose response becomes increasingly closer to the Cl-Cl one.
In fact, the increase in 95th percentile for the ensemble model average is lower than the
increase in SDII, and this is because in some models the threshold intensity in Figures 3-6,
where the sign of the change turns from negative to positive, lies beyond the 95th percentile.
When only land areas between 60°S and 60°N are taken into account (bottom panel in Table
1), the changes are generally in line with the global ones, except for the CNRM model. Over
land areas we also find changes in the highest percentiles of magnitude mostly greater than
over the globe (and thus over oceans).

We can thus conclude that the shift to a regime of more intense but less frequent

events in warmer conditions is due to the fact that precipitation intensity, especially for
intense events (beyond the 95th percentile), responds at the local level primarily to the Cl-Cl-
driven increase of water vapor amounts, while mean precipitation responds to a slower
evaporation process, driving a decrease in precipitation frequency. Noticeably, the MIROC
experiment does not appear to follow this response, i.e. in this model the increase in mean
precipitation appears to be driven by an increase in the number of light precipitation events.

While the data of Table 1 provide a diagnostic explanation of the hydroclimatic

response of Figure 7, it has also been suggested by very high resolution convection-permitting
smulations that ocean temperatures might affect the self-organization and aggregation of
convective systems (e.g. Mueller and Held 2012; Becker et al. 2017), which would also affect
the precipitation response to warming. Therefore, the study of this response might lead to a
greater understanding of the fundamental behavior of the precipitation phenomenon, and in
particular of tropical convection processes.



**3. Some consequences of the hydroclimatic response to global warming**
What are the consequences of the "more intense, less frequent" event response to
global warming illustrated in Figure 7? Obviously there can be many of them, but here we
want to provide a few illustrative examples of relevance for impact applications.
*3.1 Potential stress associated wet and dry extreme events.*
Figure 7 suggests that global warming might induce an increase in the risk of
damaging extreme wet and dry events, the former being associated with the increase in
precipitation intensity, and latter with the occurrence of longer sequences of dry days over
areas of increasing size. In order to quantify this risk, in a recent paper (Giorgi et al. 2018,
hereafter referred to as GCR18) we introduced a new index called the Cumulative
Hydroclimatic Stress Index, or CHS. In GCR18, the CHS was calculated for two types of
extreme events, the 99.9th percentile of the daily precipitation distribution (or R99.9) and the
occurrence of at least three consecutive months experiencing a precipitation deficit of
magnitude greater than 25% of the precipitation climatology for that months (or D25). Both of
these metrics thus refer to extremely wet and dry events which can be expected to produce
significant damage (see GCR18).
Taking as an example the R99.9, the CHS essentially cumulates the excess
precipitation above the 99.9th percentile threshold calculated for a given reference period (e.g.
1981-2010). Hence, the assumption is that the potential stress associated with these extremes
is proportional to the excess precipitation above the 99.9 percentile of the distribution. GCR18
calculated this quantity for a future climate projection, and then normalized it by the
corresponding value cumulated over the reference period. This normalization expresses the
potential stress due to the increase in wet extremes in Equivalent Reference Stress Years
(ERSY), where an ERSY is the mean stress per year due to the extremes during the reference



period (in our case 1981-2010). If, for example, a damage value can be associated to such
events, the ERSY can be interpreted as the mean yearly damage caused by extremes in
present climate conditions. GCR18 then carried out similar calculations for the cumulative
potential stress due to dry events by cumulating the deficit rain defined by the D25 metric. In
addition, they included exposure information within the definition of the CHS index by
multiplying the excess or deficit precipitation by future population amounts (as obtained from
Shared Socioeconomic Pathways, or SSP, Rihai et al. 2016) normalized by present day
population values. The details of these calculations can be found in GCR18.

The main results of GCR18 are summarized in Figures 8 and 9, which present maps of

the potential cumulative stress due to both dry and wet events added by climate change during
the period 2010-2100 and expressed in added ERSY (i.e. after removing the value of 90 that
would be obtained if no climate change occurred). The figures show the total ensemble-
averaged added cumulative stress for the RCP8.5 scenario without (Figure 8) and with (Figure
9) inclusion of population weighting (where the SSP5 population scenario from Rihai et al.
2016 was used).



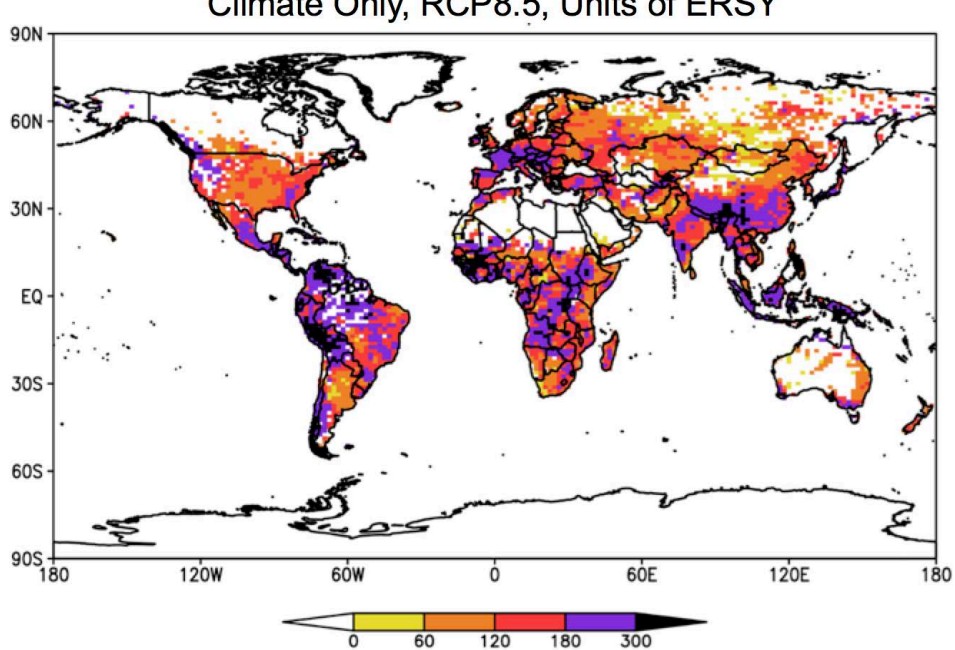

**Figure 8.** Total number of additional stress years due to increases in wet (R99.9) and dry (D25) events
for the period 2011 - 2100 including only climate variables for the RCP8.5 scenario (see text for more detail).
Units are Equivalent Reference Stress Years (ERSY) and the value does not include ERSY obtained if climate
did not change (i.e. for the period 2100 - 2011 a value of 90).





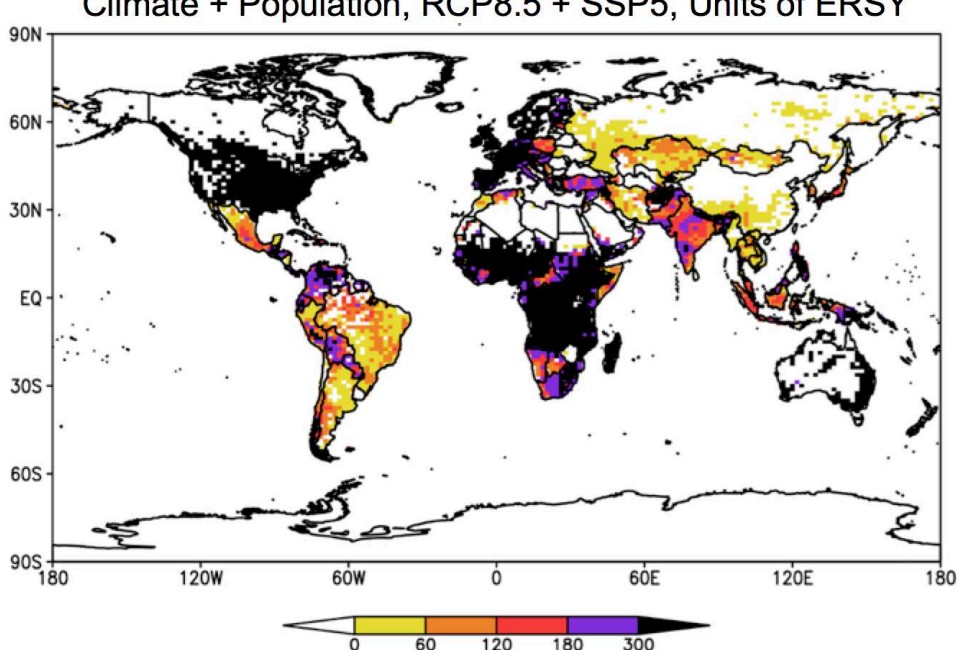

**Figure 9.** Same as Figure 8, but with the inclusion of the SSP5 population scenario (see text for more detail).

Figure 8 shows that, when only climate is accounted for, dry and wet extremes add more than 180 ERSY (and in some cases more than 300 ERSY) over extended areas of Central and South America, Europe, Western and south/central Africa, Southern and Southeastern Asia. In other words, the combined potential stress due to dry and wet extremes more than triples due to climate change by the end of the century. In this regard, GCR18 found that, when globally averaged over land regions and over all the models considered, both wet and dry extremes increased in the RCP8.5 scenario, the former adding ~120 ERSY, while the latter adding ~30 ERSY.



When population scenarios are also accounted for (Figure 9) the patterns of added
cumulative stress are considerably modified. In this case, the total number of added ERSY
exceeds 300 over the entire continental U.S. and Canada, most of Africa, Australia and areas
of South and Souteast Asia, which are projected to experience substantial population increases
in the SSP5 scenario. Conversely, we find a reduced increase in stress over East and Southeast
Asia, where population is actually projected to decrease by the end of the 21st century (see
GCR18). This result thus points to the importance of incorporating socio-economic
information in the assessment of the stress associated with climate change-driven extreme
events.
Notwithstanding the limitations and approximations of the approach of GCR18, amply
discussed in that paper, the results of Figures 8 and 9 clearly indicate that the increase of wet
and dry extremes associated with global warming can constitute a serious threat to the socio-
economic development of various regions across all continents. GCR18 also show that the
cumulative stress due to increases in extremes is drastically reduced under the RCP2.6
scenario, pointing to the importance of mitigation measures to reduce the level of global
warming.
*3.2  Impact on interannual variability.*
The interannual variability of precipitation is a key factor affecting many aspects of
agriculture and water resources and it is strongly affected by global modes of variability, such
as the El Nino Southern Oscilation (ENSO) in the tropics and the North Atlantic Oscillation
(NAO) in mid-latitudes. In this regard, the latest generation of GCM projections does not
provide strong indications concerning changes in the frequency or intensity of such modes
(e.g. IPCC 2013).



Daily and seasonal precipitation statistics are not necessarily tied, since the same
seasonal mean can be obtained via different sequences of daily precipitation events. In
addition, the intensity distribution of daily and seasonal precipitation amounts can be quite
different, the latter being often close to normal distributions (e.g. Giorgi and Coppola 2009).
On the other hand, the occurrence of longer dry spells, intensified by higher temperatures and
lower soil moisture amounts, might be expected to amplify dry seasons, while the increase in
the intensity of sequences of wet events might lead to amplified wet seasons. As a result, it
can be expected that the regime response of Figure 7 might lead to an increase in precipitation
interannual variability.
To verify this hypothesis, we calculated for the GCM ensemble of Giorgi et al.
(2014b) the change in precipitation interannual variability between future and present day 30-
year time slices using as metric the coefficient of variation (CV). The CV is defined as the (in
our case interannual) standard deviation normalized by the mean, and has been often used as a
measure of precipitation variability because it removes the strong dependence of precipitation
variability on the mean itself (Raisanen 2002; Giorgi and Bi 2005).




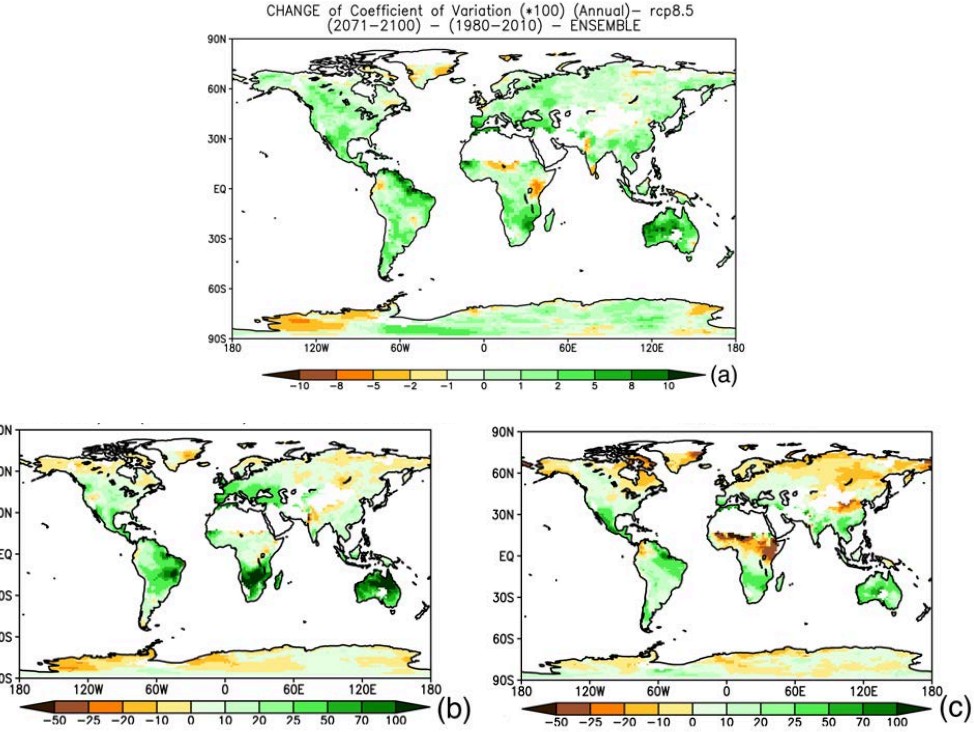

**Figure 10.** Change in precipitation interannual coefficient of variation (2071-2100 vs. 1981-2010) for a)
mean annual precipitation; b) April-September precipitation; c) October-March precipitation.

Figure 10 shows the ensemble average change in precipitation CV between the 2071-

2100 and 1981-2010 time slices for mean annual precipitation as well as precipitation
averaged over the two 6-month periods Apr-Sept and Oct-Mar. It can be seen that, when
considering annual averages, the interannual variability increases over the majority of land
areas, with exceptions over small regions scattered throughout the different continents. When
considering the two different 6-month seasons, in Apr-Sept (northern hemisphere summer,
southern hemisphere winter) variability increases largely dominate, except over areas of the
northern hemisphere high latitudes and some areas around major mountain systems. In Oct-



Mar, the areas of decreased variability are more extended over northern Eurasia, northern
North America and, interestingly, some equatorial African regions, although still the increases
are somewhat more widespread.

Although Figure 10 does not show a signal of ubiquitous sign across all land areas, it

clearly points to a prevalent increase in interannual variability associated with global
warming, at least as measured by the CV. It is important to notice that this increase occurs in
areas of both increased and decreased mean precipitation (see Figure 2), so that it is not
strongly related to the use of the CV as a metric. Finally, this result is broadly consistent with
analyses of previous generation model projections (Raisanen 2002; Giorgi and Bi 2005),
which adds robustness to this conclusion.
*3.2  Impact on precipitation predictability.*

A third issue we want to address concerns the possible effects of regime shifts on the

predictability of precipitation, an issue which has obvious implications for a number of socio-
economic activities (e.g. agriculture, hazards, tourism etc.). Indeed, precipitation is one of the
most difficult meteorological variables to forecast, since it depends on both large scale and
complex local scale processes (e.g. topographic forcing). While the chaotic nature of the
atmosphere provides a theoretical limit to weather prediction of ~10-15 days (e.g. Warner
2010), the predictability range of different types of precipitation events depends crucially on
the temporal scale of the dynamics related to the event itself. For example, the predictability
range of synoptic systems is of the order of days, while that of long-lasting weather regimes,
such as blockings, can be of weeks. It is thus clear that changes in precipitation regimes and
statistics can lead to changes in the potential predictability of precipitation.

One of the benchmark metrics that is most often used to assess the skill of a prediction

system is persistence (Warner 2010). Essentially, persistence for a lead time T assumes that a





given weather condition at a time t+T is the same as that at time t. In other words, when
applied for example to daily precipitation, it assumes that, for a lead time of N days, if day *i* is
wet (dry), day *i* + N, will also be wet (dry). The skill of a forecast system is then measured by
how much the forecast improves upon persistence. Therefore, persistence can be considered
as a "minimum potential predictability".

In order to assess whether global warming affects what we defined minimum potential

predictability for precipitation, we calculated the percentage of successfull precipitation
forecasts obtained from persistence at lead times of 1, 3 and 7 days for the 10 GCM
projections (RCP8.5) used by Giorgi et al. (2014). This percentage, calculated year-by-year
and then averaged over all land areas, is presented in Figure 11, noting that the persistence
forecast only concerns the occurrence of precipitation and not the amount.

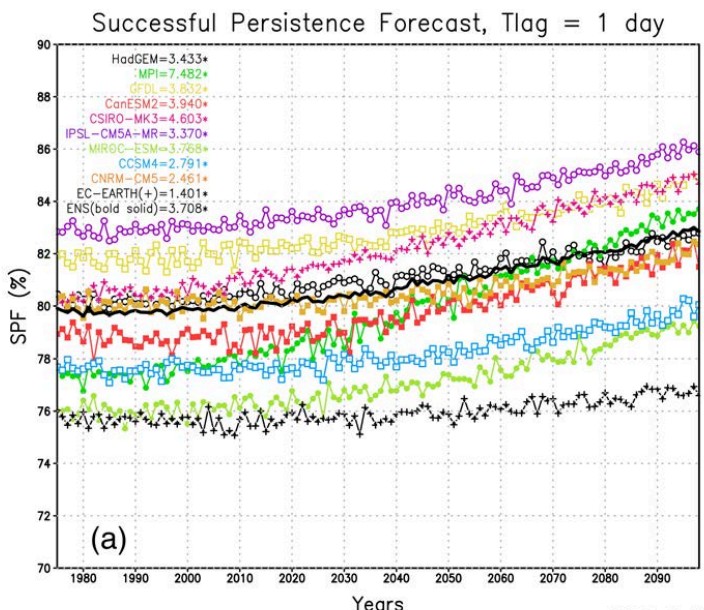



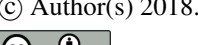


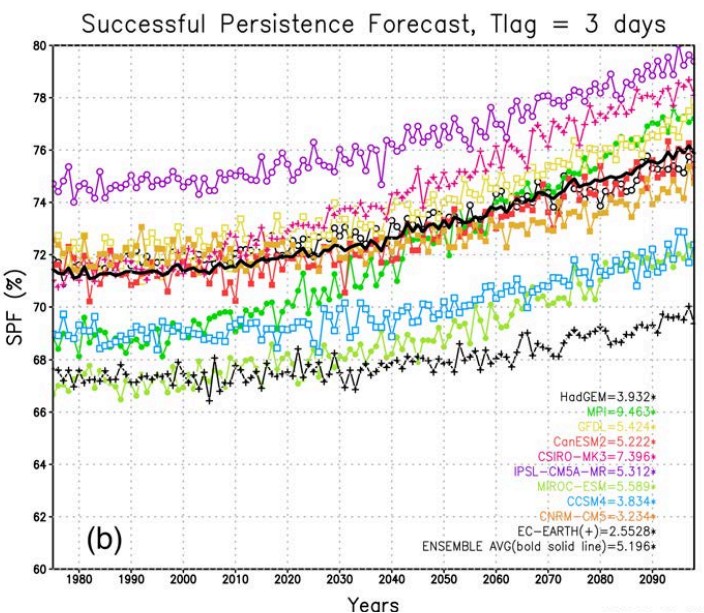


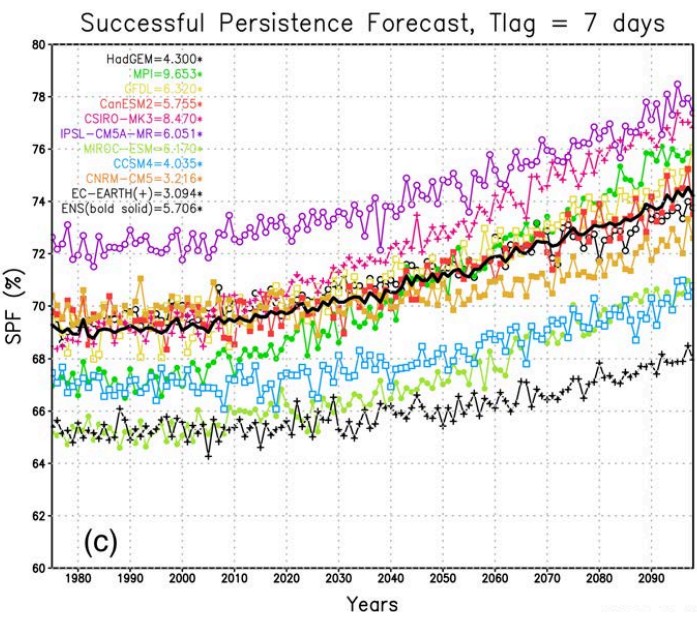


**Figure 11.** Fraction of successfull forecasts as a function of time using persistence for daily
precipitation occurrence at time lags of a) 1 day; b) 3 days; and c) 7 days, for the GCM ensemble of Giorgi et al
(2014b) (bold black line). The number in parenthesis denotes the trend in % per 100 years. Units are percentage
of days in one year for which persistence provides a successful forecast (either dry or wet).



502  Figure 11 shows that in all model projections, and thus in the ensemble averages, the

503  percent of successfull persistence forecasts increases with global warming for all three time

504  lags. This can be mostly attributed to the increase in mean dry spell length found in section 2.

505  For a lag time of 1 day, the successfull persistence forecast in the model ensemble increases

506  globally from about 80% in 2010 to about 83% in 2100, i.e. with a linear trend of ~ 3.5 %/100

507  yrs. As can be expected, the % of successfull persistence forecasts decreases with the length

508  of lag time, ~76% and 69% on 2010 for lag times of 3 and 7 days, respectively. However the

509  growth rate of this percentage also increases with lag time, 5.2%/100 yrs and 5.7%/100 yrs for

510  lag times of 3 and 7 days, respectively.

511  Despite the simplicity of the reasoning presented in this section, our results indicate

512  that global warming can indeed affect (and in our specific case, increase) the potential

513  predictability of the occurrence of dry vs. wet days. For example, persistence for the 7 day lag

514  time has the same successfull forecast rate by the middle of the 21st century as the present day

515  persistence for the 3 day lag time (~71%). Clearly, the issue of the effects of climate change

516  on weather predictability is a very complex one, with many possible implications not only

517  from the application point of view, but also for the assessment of the performance of forecast

518  systems. It is thus important that this issue is addressed with more advanced techniques and

519  metrics than we employed in our illustrative example.

520  **4. Concluding remarks**

521  In this paper we have revisited the basic responses of the characteristics of the Earth's

522  hydroclimatology to global warming through the analysis of global and regional climate

523  model projections for the 21st century. The projections examined suggested some robust

524  hydroclimatic responses, in the sense of being mostly consistent across different model





projections and being predominant over the majority of land areas. They can be summarized

as follows:

1) A decrease (increase) in the frequency of wet (dry) days

2) An increase in the mean length of dry spells

3) An increase of the mean intensity of precipitation events

4) An increase in the intensity and frequency of wet extremes

5) A decrease in the frequency of light to medium precipitation events

6) A decrease in the mean length of wet events and in the mean area covered by

precipitation

7) Occurrence of wet events of magnitude beyond that found in present climate

conditions

We discussed how this response is mostly tied to the different natures of the

precipitation and evaporation processes, and we also presented some illustrative examples of
the possible consequences of these responses, including an increase in the risks associated
with wet and dry extremes, a predominant increase in the interannual variability of
precipitation and a modification of the potential predictability of precipitation events. In
addition, some of the results 1)-7) above are consistent with previous analyses of global and
regional model projections (e.g. Tebaldi et al. 2006; Gutowski et al. 2007; Giorgi et al.
2011,2014a,b; Sillmann et al. 2013a).

Clearly, model projections indicate that the characteristics of precipitation are going to

be substantially modified by global warming, most likely to a greater extent than mean
precipitation itself. Whether these changes are already evident in the observational record is



still an open debate. Giorgi et al. (2011, 2014b) found some consistency between model
projections and observed trends in different precipitation indices for the period 1976 - 2005 in
a global and some regional observational datasets. Some indications of observed increases in
precipitation extremes over different regions of the World have also been highlighted in
different IPCC reports (IPCC 2007, 2013) and, for example, in Fischer and Knutti (2016). In
addition, data from the Munich reinsurance company suggest an increase in the occurrence of
meteorological and climatic catastrophic events, such as flood and drought, since the mid-
eighties. However, the large uncertainty and diversity in precipitation observational estimates,
most often blending in situ station observations and satellite-derived information using a
variety of methods, along with the paucity of data coverage in many regions of the World and
the large variability of precipitation, make robust statements on observed trends relatively
difficult.

A key issue concerning precipitation projections is the representation of cloud and

precipitation processes in climate models. These processes are among the most difficult to
simulate, because they are integrators of different physical phenomena and, especially for
convective precipitation, they occur at scales that are smaller than the resolution of current
GCMs and RCMs. For example, the representation of clouds and precipitation is the main
contributor to a model's climate sensitivity and the simulation of precipitation statistics is
quite sensitive to the use of different cumulus parameterizations (e.g. Flato et al. 2013). In
fact, both global and regional climate models have systematic errors in the simulation of
precipitation statistics, such as an excessive number of light precipitation events and an
underestimate of the intensity of extremes (Kharin et al. 2005; Flato et al. 2013, Sillmann et
al. 2013b). These systematic biases are related non only to the relatively coarse model
resolution, but also to inadequacies of resolvable scale and convective precipitation
parameterizations (e.g. Chen and Knutson 2008; Wehner et al. 2010; Flato et al. 2013).



Experiments with non-hydrostatic RCMs run at convection-permitting resolutions (1-3
km), in which cumulus convection schemes are not utilized and convection is explictly
resolved with non-hydrostatic wet dynamics, have shown that some characteristics of
simulated precipitation are strongly modified compared to coarser resolution models, most
noticeably the precipitation peak hourly intensity and diurnal cycle (e.g. Prein et al. 2015). It
is thus possible that some conclusions based on coarse resolution models might be modified
as more extensive experiments at convection permitting scales become available.
Despite these difficulties and uncertainties, and given the problems associated with
retrieving accurate observed estimates of mean precipitation at continental to global scales,
robust changes in different characteristics of precipitation (rather than the mean) may provide
the best opportunity to detect and attribute trends in the Earth's hydrological cycle. Moreover,
the investigation of the response of precipitation to warming may provide an important tool
towards a better understanding and modeling of key hydroclimatic processes, most noticeably
tropical convection. The ability of simulating given responses of precipitation characteristics
can also provide an important benchmark to evaluate the performance of climate models in
describing precipitation and cloud processes. Therefore, as more accurate observational
datasets become available, along with higher resolution and more comprehensive GCM and
RCM projections, the understanding of the response of the Earth's hydroclimate to global
warming, and its impacts on human societies, will continue to be one of the main research
challenges within the global change debate.
**Acknowledgements**
We thank the CMIP5 and MED-CORDEX modeling groups for making available the
simulation data used in this work, which can be found at the web site http://cmip-
pcmdi.llnl.gov/cmip5/data_portal.html and https://www.medcordex.eu/medcordex.php.    A




good portion of the material presented in this paper is drawn from the European Geosciences
Union (EGU) 2018 Alexander von Humboldt medal lecture delivered by one of the authors
(F.G.).

**Competing Financial Interests**
The authors declare no competing financial interests.

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
