# Peer review of "The response of precipitation characteristics to global warming from climate projections"

_Earth System Dynamics, 2018_

## Referee Comment (RC1) · Anonymous Referee #1 · 30 Nov 2018

The manuscript by Giorgi et al analyze climate model simulations – mostly global, but also two regional models – describing and analyzing sundry characteristics of the hydrologic cycle and how they are projected to respond to warming. They include a short comparison with regional climate simulations, and then go on to discuss some consequences of the changes: a measure of hydroclimate stress, interannual variability, and potential predictability change.

While the response of precipitation to warming is certainly an important question within the scientific scope of the journal, the majority of this study presents concepts and ideas that are very much established in the existing literature, but through analysis that

is less robust or comprehensive than other studies, and meanwhile the study does not give credit or acknowledgement to many of these existing studies. As a result, the original contribution of this work is not clear. It is stated in the introduction that the intention is not to "provide a review of the extensive literature on this topic," and also in the conclusion that "some of the results . . . are consistent with previous analyses." But statements are made which contradict existing work, and analysis presented here falls short of the standards established by this other other work. In some cases, sections of the paper are undermined when considered in light of the existing, unacknowledged literature. Overall, it seems to me that the degree to which existing literature is neglected and to which the conclusions lack originality is problematic. The only part of the study that is both novel (to my knowledge) and convincing is section 3.2, on potential predictability, but this part of the study is more rough than others, and gets only a passing mention in the conclusions.

Specific comments:

Line (L) 57-58: It seems that using a threshold of 1 mm/d applied to climate models could be problematic. This threshold makes sense at a rain gauge, but it means something different averaged over a model grid cell – and what it means will vary with the resolution of the model (see e.g. Chen and Dai 2018).

L91: There is a substantial literature that explores robust responses of the hydrologic cycle, and restricting analysis to just 1 or 10 of the models is an outdated approach.

L121-57: The "rich get richer" aka "wet get wetter" paradigm has been shown not to hold over land by Byrne and O'Gorman (2015), which would point toward it being somewhat irrelevant for the consequences discussed in section 3.

L132-135: These descriptions of the changes in ITCZ and monsoons do not reflect the current state of understanding. See, for example, a review by Byrne et al., (2018) that includes changes in ITCZ strength, and Biasutti et al., (2018) on changing monsoon strength.

L170, 226-227, Other studies have examined the changing distribution of precipitation and include uncertainty estimates and also compare across models. Pendergrass and Hartmann (2014) examine over 20 models for RCP8.5 and CO2 increase scenarios, and show that the MPI family of models has a very different behavior in terms of its heavy precipitation response to warming, mostly in the tropics, compared to other climate models, which calls into question how robust the results shown here would be across models.

L228-238: Recent work by Thackeray et al., (2018) shows compensation between extreme and non-extreme events across CMIP5 RCP8.5 simulations from different models. This highlights the role of the energy budget in affecting the distribution of precipitation – but regional climate models don't capture these energetic feedbacks.

Figure 7: This figure is difficult to understand. Some attempt should be made to label what the boxes mean on the figure, visually, and/or axis labels should be included. The lack of labeling of the figure combined with the way the index labels are included is confusing. What is the purpose of the red arrows? Doesn't R95 relate to only some of the boxes (the bigger ones), rather than all of them, like SDII does? Doesn't HY-INT relate to all of the precipitation events, rather than just a few?

L296, L329-332, L339: Extreme precipitation responds not just to changes in moisture, but also to changes in circulation. In some models the increases in the most extreme precipitation are substantially larger than Clausius-Clapeyron, and in some regions they are much smaller. The regional variation in extremes is documented by Pfahl et al., (2017). The variation in the warming response to percentile definitions, including the range across models extending well above CC, is shown in Pendergrass and Hartmann (2014).

Figure 8 and 9: These maps seem to show the sum of the cumulative stress from wet and dry extremes that are shown in Fig 2 of GCR18, though this is not explicitly stated. In that figure of GCR18, wet extremes drive positive stress (ERSY), while dry

extremes sometimes drive positive stress and other times drive negative stress. In locations where dry extremes drive increasing stress, adding ESRY from wet and dry extremes results in cancellation, which is effectively an assumption that wet extremes will mitigate dry extremes. This cancellation does not seem justified; a short timescale heavy precipitation event could still cause flooding in situations where there has been a deficit of precipitation on a longer timescale, depending on the characteristics of the surface where it falls. It is worrisome that this section consists of an incremental and not-well-justified advance on the author's recent work.

L407, Figure 9 caption: The figure caption says that the text includes more detail on how population is incorporated, but the text only seems to say "population scenarios are also accounted for." Please describe how population is accounted for. Furthermore, GCR18 was not the first study to include population weighting of trends in the hydrologic cycle, but none of the previous work is acknowledged here or in GCR18 – one example is Sedláček and Knutti (2014).

L426-9: "the latest generation of GCM projections does not provide strong indications concerning changes in the frequency or intensity of such modes [ENSO and NAO are explicitly stated]." There is substantial literature documenting the effects of changes in ENSO, in particular, on interannual variability of precipitation. One example is Power et al., (2013).

L424-467: The effect of increasing variability in the CMIP5 ensemble on a variety of timescales including daily, interannual, and across multiple years, was documented by Pendergrass et al., (2017).

L521-543: As mentioned in the general comments, it seems that the only novel conclusion arrived at here may be the potential predictability.

L562-578: I certainly agree with the statement that conclusions based on coarse resolution models will have to be revisited and potentially modified as our ability to simulate precipitation advances. But as mentioned in my comment regarding lines 228-238, recent work by Thackeray et al., (2018) shows the key role that global energy constraints play in determining how the distribution of precipitation responds to warming – a factor that is not accounted for in regional climate models. As written, this paragraph can be interpreted as implying that high resolution regional climate models are a solution, whereas the findings of Thackeray et al., (2018) provide evidence to the contrary. This drawback of regional climate modeling should also be acknowledged.

Typos and minor comments

L260: MDSL is not defined.

L355: "stress associated wet" should be "stress associated with wet"

L575: "strongly" should be "strongly"

Additional references

Biasutti, M., Voigt, A., Boos, W. R., Braconnot, P., Hargreaves, J. C., Harrison, S. P., Kang, S. M., Mapes, B. E., Scheff, J., Schumacher, C., Sobel, A. H. and Xie, S.-P.: Global energetics and local physics as drivers of past, present and future monsoons, Nat. Geosci., 11(6), 392–400, doi:10.1038/s41561-018-0137-1, 2018.

Byrne, M. P. and O'Gorman, P. A.: The response of precipitation minus evapotranspiration to climate warming: Why the "Wet-get-wetter, dry-get-drier" scaling does not hold over land, J. Clim., 28(20), 8078–8092, doi:10.1175/JCLI-D-15-0369.1, 2015.

Byrne, M. P., Pendergrass, A. G., Rapp, A. D. and Wodzicki, K. R.: Response of the Intertropical Convergence Zone to Climate Change: Location, Width, and Strength, Curr. Clim. Chang. Reports, 4(4), 355–370, doi:10.1007/s40641-018-0110-5, 2018.

Chen, D. and Dai, A.: Dependence of estimated precipitation frequency and intensity on data resolution, Clim. Dyn., 50(9–10), 3625–3647, doi:10.1007/s00382-017-3830-7, 2018.

Pendergrass, A. G. and Hartmann, D. L.: Changes in the distribution of rain frequency and intensity in response to global warming, J. Clim., 27(22), 8372–8383, doi:10.1175/JCLI-D-14-00183.1, 2014.

Pendergrass, A. G., Knutti, R., Lehner, F., Deser, C. and Sanderson, B. M.: Precipitation variability increases in a warmer climate, Sci. Rep., 7(1), 17966, doi:10.1038/s41598-017-17966-y, 2017.

Pfahl, S., O'Gorman, P. A. and Fischer, E. M.: Understanding the regional pattern of projected future changes in extreme precipitation, Nat. Clim. Chang., 7(6), 423–427, doi:10.1038/nclimate3287, 2017.

Power, S., Delage, F., Chung, C., Kociuba, G. and Keay, K.: Robust twenty-first-century projections of El Niño and related precipitation variability., Nature, 502(7472), 541–5, doi:10.1038/nature12580, 2013.

Sedláček, J. and Knutti, R.: Half of the world's population experience robust changes in the water cycle for a 2 '°C warmer world, Environ. Res. Lett., 9(4), 044008, doi:10.1088/1748-9326/9/4/044008, 2014.

Thackeray, C. W., DeAngelis, A. M., Hall, A., Swain, D. L. and Qu, X.: On the Connection Between Global Hydrologic Sensitivity and Regional Wet Extremes, Geophys. Res. Lett., doi:10.1029/2018GL079698, 2018.

---

## Referee Comment (RC2) · Anonymous Referee #2 · 1 Dec 2018

The manuscript has a very important and exciting objective but in the end, it looks like a very routine analysis and I am unable to find any scientific merit. It may be because of the writing style or maybe because the analysis looks very standard. I am also unable to find any new methodological development. The conclusions are as per intuitions and it suits better as a review or assessment article. The hydrological simulations also need a detailed description. The human component is also a major component in hydrology and I am not sure how the authors are incorporating the same.

---

## Author Comment (AC1) · 21 Dec 2018

Response to the issue of novelty raised by both reviewers.

First, we would like to thank the reviewers for their comments, which are certainly well taken. Both reviewers question the novelty of the paper, so perhaps we should clarify the genesis of this article. The paper is essentially drawn from the Alexander von Humboldt's medal lecture given by the first author (FG) at the 2018 EGU general assembly. The paper was solicited by the EGU office, since it is apparently standard procedure that medal recipients are asked to synthesize their lecture into a paper for an EGU journal. As such, the paper is not intended to present a review of the large

body of literature covering this field. By the same token, it is not intended to be a paper containing new research, since the medal lecture was mostly a synthesis of the author team contribution to this topic. In other words, the paper mostly synthesizes the main messages that came out of previous work conduted by the author team, plus some limited new analyses. Having said this, we want to clarify a few points. First, none of the figures of this paper have been taken directly from previous papers. For the medal lecture and for this paper we produced figures that, although mostly not the result of a conceptually new analysis, had not been presented previously and were produced specifically for the medal lecture. Second, we certainly acknowledge the fact that the paper probably does not cite many relevant previous studies, however this comes directly from the fact that the paper is not a review, but mostly a synthesis of previous work by the authors. Certainly this does not come from a willingness to ignore previous work. We tried to amend this problem by including the literature cited by Referee 1, plus other papers that we considered most relevant for the text, although certainly it is possible that important papers are still not cited. As the referee mentions, there is a huge body of work on this topic, and it is difficult to keep track of it all. Finally, although it is certainly possible that some outcomes of our analysis might not be entirely in line with previous work, we do not think that they "contradict" previous findings, as we hope to show in the response to the specific comments. We hope we have clarified the origin and nature of this paper and it is clearly an editorial decision whether a paper of such nature belongs to ESD, which we deemed was the most appropriate EGU journal to submit it to.

Response to specific comments.

Referee 1.

First, we would like to thank the referee for his/her detailed review, for his/her suggestions and for pointing us to some recent literature which we were not aware of. Line (L) 57-58: It seems that using a threshold of 1 mm/d applied to climate models could be problematic. This threshold makes sense at a rain gauge, but it means something dif-

ferent averaged over a model grid cell – and what it means will vary with the resolution of the model (see e.g. Chen and Dai 2018). Reply: This comment is well taken, however in all our previous papers we used a threshold of 1 mm/day consistently for both gridded observations and models (to avoid the issue of continuous drizzle events), as has been often done in this type of analysis, at least to our knowledge. To use now a different threshold would make the consistency with our previous studies more difficult, and especially for such a synthesis paper, we do not think it would be advisable. In addition, when directly intercomparing model data or comparing models and observed data we always interpolated the data onto common grids. Anyways, we now mention this issue in the revised text. Revision: We added the sentence on lines 209-215: "Note that, as in our previous work (Giorgi et al. 2014b), throughout this paper a rainy day is considered has having a precipitation amount of at least 1 mm/day, so that drizzle days are removed. In this regard, the choice of a precipitation threshold to define a rainy day makes the calculation of precipitation frequency and intensity dependent on the resolution of the data (e.g. Chen and Dai 2018). This issue should be taken into account when analyzing precipitation statistics and here, as well as in previous work, we conduct direct cross model or data-model intercomparisons only after having interpolated the data onto common grids.

L91: There is a substantial literature that explores robust responses of the hydrologic cycle, and restricting analysis to just 1 or 10 of the models is an outdated approach. Reply: As mentioned, we mostly base our analysis on the 10 models mentioned in the text, mostly because these are the models used in previous work assessing daily precipitation statistics (Giorgi et al. 2014b). These 10 models were chosen because they were the only ones among the full CMIP5 dataset for which daily data were available at the time we did our previous study. Besides the fact that this sub-ensemble includes some of the most commonly used models, some analysis of mean and seasonal data showed that this sub-ensemble behaves quite similarly to the full CMIP5 one. Although we realize that this conclusion cannot be automatically extended to daily statistics, we still feel that this 10-GCM ensemble is at least qualitatively representative of the full
CMIP5 (especially given the almost full consistency we find across the 10 models, see Table 1). Concerning the RCM used, this is our own RCM and we added this figure to the paper just to provide an illustrative example of how such type of result can be found also in RCMs. We have also looked at a few other RCM results from EURO-CORDEX (e.g. REMO and RACMO, see figures 1-2 and 3-4) and found a qualitatively similar behavior, however we did not include additional figures in the paper for brevity. To be honest, based on all the analysis we have done showing strong model consistency, we do not think that the conclusions of our paper would change significantly when using larger ensembles. Since, however, we only show results from one RCM, we realize that it is not justified to mention RCMs in the title, and therefore we modified it.

Revision: We changed the title of the paper to: The response of precipitation characteristics to global warming from climate projections.

L121-57: The "rich get richer" aka "wet get wetter" paradigm has been shown not to hold over land by Byrne and O'Gorman (2015), which would point toward it being somewhat irrelevant for the consequences discussed in section 3.

Reply: Point well taken. Perhaps we used the "rich get richer / poor get poorer" analogy too casually. We intended it in the general "IPCC jargon" of wet areas getting predominantly more precipitation (mid to high latitudes) and dry areas getting predominantly less precipitation (e.g. Mediterranean, South west US, Southern Africa, southern S. America etc.). We are aware of the paper mentioned by the referee showing that a more strict analysis of P-E over land technically does not support this type of statement (because of the E part of it), therefore since this may evidently cause an ambigous interpretation in terms of hydrologic balance, we have removed reference to this sentence and modified the related text. Revision: We removed the paragraph " The patterns of Figure 2 have been often referred to as "the rich get richer and the poor get poorer" ... are projected to become drier." We changed the sentence "As a result of all these processes it is thus possible that the "rich get richer, poor get poorer" patterns might be significantly modified as we move to substantially higher resolution

models." into " As a result of all these processes it is thus possible that the large scale precipitation change patterns of Fig. 2 might be significantly modified as we move to substantially higher resolution models". (L178-180). We changed the sentence " On the other hand, the question could be posed: "How richer will the rich get and how poorer will the poor get?". into " On the other hand, a key question concerning the precipitation response to global warming is: "How will precipitation change patterns affect different socioeconomic sectors?" ". (L181-182).

L132-135: These descriptions of the changes in ITCZ and monsoons do not reflect the current state of understanding. See, for example, a review by Byrne et al., (2018) that includes changes in ITCZ strength, and Biasutti et al., (2018) on changing monsoon strength. Reply: We thank the referee for point us to these recent review papers. Our statement was based mostly on the outcome of the last IPCC report (2013), and we have revised it based on the conclusions of these two papers, keeping in mind that this is not the focus of our study. Revision: We substituted the sentence " Conversely, the increase in precipitation over the ITCZ is due to increased evaporation over the equatorial oceans, which feeds and intensifies local convective systems. Finally, over monsoon regions, a general increase of precipitation has been attributed to the greater water-holding capacity of the atmosphere that counterbalances a decrease in monsoon circulation strength (IPCC 2013)." With "The Intertropical Convergence Zone (ITCZ) shows narrowing and greater precipitation intensity, especially in the core of the Pacific ITCZ, associated with increased organized deep convective activity towards the ITCZ center and decreased activity along its edges (Byrne et al. 2018). Finally, over monsoon regions, a general increase of precipitation has been attributed to the greater water-holding capacity of the atmosphere counterbalancing a decrease in monsoon circulation strength (IPCC 2013), however more detailed analyses of how global constraints on energy and momentum budgets affect regional scale circulations are needed for a better understanding of the monsoon response to global warming (Biasutti et al. 2018)." (L135-143)

L170, 226-227, Other studies have examined the changing distribution of precipitation and include uncertainty estimates and also compare across models. Pendergrass and Hartmann (2014) examine over 20 models for RCP8.5 and CO2 increase scenarios, and show that the MPI family of models has a very different behavior in terms of its heavy precipitation response to warming, mostly in the tropics, compared to other cli- mate models, which calls into question how robust the results shown here would be across models. Reply: Thanks for pointing us to the Pendergrass and Hartmann (2014) (PH14) paper, which as far as we can see, when dealing with different responses across individual models, compares the MPI, GFDL and IPSL models, showing that indeed the MPI behaves differently from the other two in terms of extreme events. Note, in this regard, that PH14 appears to indicate that the behavior of the GFDL and IPSL models for very high intensity events may be related to the occurrence of grid point storms in the non-convective component of precipitation, a behavior clearly spurious. Similarly to PH14, we also find that the MPI hydrologic cycle is one of the most responsive to warming across the 10 CMIP5 models we analysed, but qualitatively in line with most of the others (see Table 1). You can see, as examples, the response plots for HadGEM and EC-EARTH (Figure 5-6 and Figure 7-8), which are consistent with the MPI one (Figure 3-4 of the paper). In our experience, if anything, the GFDL model shows more anomalous responses than MPI (see Table 1), and we would add that in all our analysis of precipitation in CMIP5 models, the MPI consistently scored among the best, and definitely better than GFDL. Indeed, the reason we show the MPI model results is that they are the best illustrative ones of a common behavior, and from a model that we think is among the best in reproducing the observed hydrologic cyle. Moreover, we also find that our basic conclusions are generally in line with those of Pendergrass and Hartmann, so we would prefer to leave the MPI model results in the paper.

Revision:. No revision necessary, but we now cite the Pendergrass and Hartmann paper in the introduction, where we list previous studies.

L228-238: Recent work by Thackeray et al., (2018) shows compensation between extreme and non-extreme events across CMIP5 RCP8.5 simulations from different models. This highlights the role of the energy budget in affecting the distribution of precipitation but regional climate models don't capture these energetic feedbacks. L562-578: I certainly agree with the statement that conclusions based on coarse resolution models will have to be revisited and potentially modified as our ability to simulate precipitation advances. But as mentioned in my comment regarding lines 228-238, recent work by Thackeray et al., (2018) shows the key role that global energy constraints play in determining how the distribution of precipitation responds to warming – a factor that is not accounted for in regional climate models. As written, this paragraph can be interpreted as implying that high resolution regional climate models are a solution, whereas the findings of Thackeray et al., (2018) provide evidence to the contrary. This drawback of regional climate modeling should also be acknowledged.

Reply: On these two points we do not agree with the referee. We read carefully the paper by Thackeray et al. (2018) and do not think it supports an overarching statement to the effect that RCMs cannot be used to study the issue of how precipitation intensity responds to warming (neither the Thackeray et al. paper directly suggests such conclusion). The response of regional precipitation responses to changes in global energy budgets, for example through atmospheric rivers or changes in large scale monsoon flow, is accounted for by the driving GCM, which then transmits this information to the RCMs through the boundary conditions. The role of the RCM is not to capture or modify this large scale response but only to add fine scale detail to the information provided by the GCM, accounting for local processes (e.g. topography) and local circulations, energy and water fluxes. The higher resolution of RCMs can indeed improve the simulation of individual events, or the organizatin of convective events, and thus it can actually help in this aspect of the problem. An overarching statement on drawbacks by RCMs would have to be supported by specific experiments (not present in the Thackeray et al. paper, which in fact, as mentioned, does not include any statement about RCMs). If anything, we show that the results from RCMs are qualitatively in line with

those of GCMs. We have anyways included reference to the work of Thackeray et al. (2018) in the introduction. Revision: The Thackeray et al. paper is now cited in the following paragraph of Section 2.1 " Although precipitation increases globally, at the regional level we can find relatively complex patterns of change, with areas of increased and areas of decreased precipitation. These patterns are closely related to changes in global circulation features, global energy and momentum budgets, local forcings (e.g. topography, land use) and energy and water fluxes affecting convective activity (e.g. Thackeray et al. 2018; Byrne et al. 2018). ". (L120-122)

Figure 7: This figure is difficult to understand. Some attempt should be made to label what the boxes mean on the figure, visually, and/or axis labels should be included. The lack of labeling of the figure combined with the way the index labels are included is confusing. What is the purpose of the red arrows? Doesn't R95 relate to only some of the boxes (the bigger ones), rather than all of them, like SDII does? Doesn't HY-INT relate to all of the precipitation events, rather than just a few? Reply: We take the point that this figure is difficult to read without a proper detailed explanation, which however is really not necessary, since the text is quite self-exaplnatory. We have thus decided to remove the figure. Revision: We took out Figure 7, and in its place we have introduced the acronym Higher Intensity Reduced Frequency (HIRF) response.

L296, L329-332, L339: Extreme precipitation responds not just to changes in moisture, but also to changes in circulation. In some models the increases in the most extreme precipitation are substantially larger than Clausius-Clapeyron, and in some regions they are much smaller. The regional variation in extremes is documented by Pfahl et al., (2017). The variation in the warming response to percentile definitions, including the range across models extending well above CC, is shown in Pendergrass and Hartmann (2014). Reply: Of course we agree with the referee that local circulations and thermodynamical effects modulate the CC response, in fact on average the models do not have a CC response (see Table 1) and the response itself changes for different precipitation percentile categories. We thought this was implicit in our
statements, but evidently not, so we have expanded these comments along the lines indicated by the referee. Revision: We replaced the original paragraph "An explanation for the hydroclimatic response to global warming illustrated in Figure 7 is related to the fact that, on the one hand, the mean global precipitation change roughly follows the mean global evaporation increase, i.e. 1.5-2.0 %/DGW (Trenberth et al. 2007, Figure 1), while, on the other hand the intensity of precipitation, in particular for high and extreme precipitation events, is more tied to the increase in the water holding capacity of the atmosphere, which is in turn regulated by the Clausius-Clapeyron (Cl-Cl) response of about 7%/DGW (e.g. Trenberth et al. 2003; Pall et al. 2007; Lenderink and van Meijgaard 2008; Chou et al. 2012; Singleton and Toumi 2013; Ivancic and Shaw 2016; Fischer and Knutti 2016). Therefore the increase in precipitation intensity can be expected to be larger than the increase in mean precipitation, which implies a decrease in precipitation frequency." With "An explanation for the HIRF hydroclimatic response to global warming is related to the fact that, on the one hand, the mean global precipitation change roughly follows the mean global evaporation increase, i.e. 1.5-2.0 %/DGW (Trenberth et al. 2007, Figure 1). On the other hand, the intensity of precipitation, in particular for high and extreme precipitation events, is more tied to the increase in the water holding capacity of the atmosphere, which is in turn regulated by the Clausius-Clapeyron (Cl-Cl) response of about 7%/DGW, although the precipitation response is modulated by regional and local circulations, along with energy and water fluxes, which might lead to super- or sub- Cl-Cl responses (e.g. Trenberth et al. 2003; Pall et al. 2007; Lenderink and van Meijgaard 2008; Giorgi et al, 2011; Chou et al. 2012; Singleton and Toumi 2013; Pendergrass and Hartmann 2014; Ivancic and Shaw 2016; Fischer and Knutti 2016; Pfahl et al. 2017). Therefore the increase in precipitation intensity can be expected to be generally larger than the increase in mean precipitation, which implies a decrease in precipitation frequency." (L369-381)

Figure 8 and 9: These maps seem to show the sum of the cumulative stress from wet and dry extremes that are shown in Fig 2 of GCR18, though this is not explicitly stated. In that figure of GCR18, wet extremes drive positive stress (ERSY), while dry
extremes sometimes drive positive stress and other times drive negative stress. In locations where dry extremes drive increasing stress, adding ESRY from wet and dry extremes results in cancellation, which is effectively an assumption that wet extremes will mitigate dry extremes. This cancellation does not seem justified; a short timescale heavy precipitation event could still cause flooding in situations where there has been a deficit of precipitation on a longer timescale, depending on the characteristics of the surface where it falls. It is worrisome that this section consists of an incremental and not-well-justified advance on the author's recent work. Reply: We think that the referee here has misunderstood how we calculate the sum of the stress from wet and dry extremes, or perhaps that we did not explain it well. Here we calculate the stress for wet extremes and dry extremes separately, and then we add them. Therefore there is no cancellation of stress if, say a wet extreme is followed by a dry extreme, which would be the case if we calculated them together. In other words, we are just plotting the values found in figure 4 of the original paper in a different way (as a map rather than regional averages) and without modifying the calculations. We are now explaining this more clearly. Revision: We added the sentence " The values in the figures were computed by first calculating the stress contribution in ERSY of wet and dry extremes separately and then adding them, so that there is no cancellation of stress if, say, a wet extreme is followed by a dry extreme." (L484-486). L407, Figure 9 caption: The figure caption says that the text includes more detail on how population is incorporated, but the text only seems to say "population scenarios are also accounted for." Please describe how population is accounted for. Further- more, GCR18 was not the first study to include population weighting of trends in the hydrologic cycle, but none of the previous work is acknowledged here or in GCR18 – one example is SedlácĔĞek and Knutti (2014). Reply: The way population amounts is accounted for is explained on lines 378-382 of the original manuscript " In addition, they included exposure information within the definition of the CHS index by multiplying the excess or deficit precipitation by future population amounts (as obtained from Shared Socioeconomic Pathways, or SSP, Ri- hai et al. 2016) normalized by present day population values. The details of these

calculations can be found in GCR18." Also, we do not think that we claim either here on in the previous paper by Giorgi et al. (2018) to be the first to include population information in these types of calculations. In fact we essentially followed an approach already used in a previous paper co-authored by one of the authors (Diffenbaugh et al. 2007), which, incidentally, was not referenced for example by Sedlacek and Knutti (2014) (these things happen). Anyways, we added reference to the paper by Sedlacek and Knutti, as well as the paper by Diffenbaugh et al. (2007). Revision: The sentence above has been changed to "In addition, similarly to Diffenbaugh et al. (2007) and Sedlacek and Knutti (2014), they included exposure information within the definition of the CHS index by multiplying the excess or deficit precipitation by future population amounts (as obtained from Shared Socioeconomic Pathways, or SSP, Rihai et al. 2016) normalized by present day population values. The details of these calculations can be found in GCR18". (L473-477)

L426-9: "the latest generation of GCM projections does not provide strong indications concerning changes in the frequency or intensity of such modes [ENSO and NAO are explicitly stated]." There is substantial literature documenting the effects of changes in ENSO, in particular, on interannual variability of precipitation. One example is Power et al., (2013). Reply: Our statement about changes of ENSO and NAO related to global warming is based on the latest IPCC report (2013), where no robust conclusions are found from global model projections on changes in these modes (e.g. increase of decrease in the magnitude or structure of ENSO events, see SPM 2013, page 23). The Power et al. paper also cites a few other publications mentioning this large uncertainty in ENSO response to warming. Our understanding of this debate is that it is still relatively open, and perhaps the new generation of CMIP6 projections will shed new light into it. We however acknowledge the comment by the referee and have accordingly modified this sentence. Revision: The sentence " In this regard, the latest generation of GCM projections does not provide strong indications concerning changes in the frequency or intensity of such modes (e.g. IPCC 2013)" was modified in " In this regard, the latest generation of GCM projections does not provide definitive indications

concerning changes in the frequency or intensity of such modes (e.g. IPCC 2013), although some works suggest the presence of robust changes in projected spatial patterns of ENSO-driven precipitation and temperature variability (e.g. Power et al. 2013)". (L531-535) L424-467: The effect of increasing variability in the CMIP5 ensemble on a variety of timescales including daily, interannual, and across multiple years, was documented by Pendergrass et al., (2017). Revision: We added this reference.

L521-543: As mentioned in the general comments, it seems that the only novel conclusion arrived at here may be the potential predictability. Reply: Again, the purpose of the paper was not to present novel results thorughout. This is the only case of a "new" application.

Typos and minor comments have been corrected, and the additional references included.

[Figure]

**Fig. 1.** Ratio of future to reference normalized frequency of daily precipitation intensity for the three future time slices for the EURO-CORDEX Domain using the REMO RCM driven by global fields from MPI GCM.

[Figure]

**Fig. 2.** A zoom of Figure 1

**RACMO rcp8.5 - EURO Domain - Land Only**

Legend:
- 2071-2100 (black)
- 2041-2070 (red)
- 2011-2040 (green)

**Fig. 3.** Same as Figure 1 but for RACMO RCM driven by EC-EARTH GCM.

**Fig. 4.** Same as Figure2 but for RACMO RCM driven by EC-EARTH GCM.

[Figure]

**Fig. 5.** Same as Figure 1 but for HadGEM GCM and for tropical land areas.

[Figure]

**Fig. 6.** Same as Figure 2 but for HadGEM GCM and for tropical land areas.

[Figure]

**Fig. 7.** Same as Figure 1 but for EC-EARTH GCM and for tropical land areas.

[Figure]

**Fig. 8.** Same as Figure 2 but for EC-EARTH GCM and for tropical land areas.

---

## Author Comment (AC2) · 21 Dec 2018

Response to the issue of novelty raised by both reviewers. First, we would like to thank the reviewers for their comments, which are certainly well taken. Both reviewers question the novelty of the paper, so perhaps we should clarify the genesis of this article. The paper is essentially drawn from the Alexander von Humboldt's medal lecture given by the first author (FG) at the 2018 EGU general assembly. The paper was solicited by the EGU office, since it is apparently standard procedure that medal recipients are asked to synthesize their lecture into a paper for an EGU journal. As such, the paper is not intended to present a review of the large body of literature covering this field. By

the same token, it is not intended to be a paper containing new research, since the medal lecture was mostly a synthesis of the author team contribution to this topic. In other words, the paper mostly synthesizes the main messages that came out of previous work conduted by the author team, plus some limited new analyses. Having said this, we want to clarify a few points. First, none of the figures of this paper have been taken directly from previous papers. For the medal lecture and for this paper we produced figures that, although mostly not the result of a conceptually new analysis, had not been presented previously and were produced specifically for the medal lecture. Second, we certainly acknowledge the fact that the paper probably does not cite many relevant previous studies, however this comes directly from the fact that the paper is not a review, but mostly a synthesis of previous work by the authors. Certainly this does not come from a willingness to ignore previous work. We tried to amend this problem by including the literature cited by Referee 1, plus other papers that we considered most relevant for the text, although certainly it is possible that important papers are still not cited. As the referee mentions, there is a huge body of work on this topic, and it is difficult to keep track of it all. Finally, although it is certainly possible that some outcomes of our analysis might not be entirely in line with previous work, we do not think that they "contradict" previous findings, as we hope to show in the response to the specific comments. We hope we have clarified the origin and nature of this paper and it is clearly an editorial decision whether a paper of such nature belongs to ESD, which we deemed was the most appropriate EGU journal to submit it to.

Response to specific comments.

Referee 2 The manuscript has a very important and exciting objective but in the end, it looks like a very routine analysis and I am unable to find any scientific merit. It may be because of the writing style or maybe because the analysis looks very standard. I am also unable to find any new methodological development. The conclusions are as per intuitions and it suits better as a review or assessment article. The hydrological simulations also need a detailed description. The human component is also a major

component in hydrology and I am not sure how the authors are incorporating the same.

Reply: Concerning the novelty of the approach, see our response above.

Concerning the hydrological calculations, we are not sure what the refereee is referring to, since the are no hydrological simulations included in the paper.

―――――――――――――――